# Plant-Derived as Alternatives to Animal-Derived Bioactive Peptides: A Review of the Preparation, Bioactivities, Structure–Activity Relationships, and Applications in Chronic Diseases

**DOI:** 10.3390/nu16193277

**Published:** 2024-09-27

**Authors:** Li Sun, Jinze Liu, Zhongmei He, Rui Du

**Affiliations:** 1College of Chinese Medicinal Materials, Jilin Agricultural University, Changchun 130118, China; 20221595@mails.jlau.edu.cn (L.S.); liujinze0602@mails.jlau.edu.cn (J.L.); 2Jilin Provincial Engineering Research Center for Efficient Breeding and Product Development of Sika Deer, Changchun 130118, China

**Keywords:** plant-derived bioactive peptides, enzymolysis, bioactivities, structure–activity relationship, chronic diseases

## Abstract

**Background/Objectives:** At present, a large number of bioactive peptides have been found from plant sources with potential applications for the prevention of chronic diseases. By promoting plant-derived bioactive peptides (PDBPs), we can reduce dependence on animals, reduce greenhouse gas emissions, and protect the ecological environment. **Methods:** In this review, we summarize recent advances in sustainably sourced PDBPs in terms of preparation methods, biological activity, structure–activity relationships, and their use in chronic diseases. **Results:** Firstly, the current preparation methods of PDBPs were summarized, and the advantages and disadvantages of enzymatic method and microbial fermentation method were introduced. Secondly, the biological activities of PDBPs that have been explored are summarized, including antioxidant, antibacterial, anticancer and antihypertensive activities. Finally, based on the biological activity, the structure–activity relationship of PDBPs and its application in chronic diseases were discussed. All these provide the foundation for the development of PDBPs. However, the study of PDBPs still has some limitations. **Conclusions:** Overall, PDBPs is a good candidate for the prevention and treatment of chronic diseases in humans. This work provides important information for exploring the source of PDBPs, optimizing its biological activity, and accurately designing functional foods or drugs.

## 1. Introduction

With increased awareness of environmental pollution, animal cruelty, and the negative health effects of animal-based foods, consumers are turning to plant-based vegan alternatives [1]. The EAT-Lancet Commission report also points to a shift towards more plant-based diets to meet the UN Sustainable Development Goals and the Paris Agreement, as well as using the Earth Healthy Diet as a reference for healthy and sustainable diets for a growing population [2]. In recent decades, plant-derived bioactive peptides (PDBPs) have attracted much attention due to their low cost and environmental sustainability. The use of plants as ingredients in food and other industries is widely accepted worldwide, so there is no religious or sociocultural prejudice against their use [3]. In addition, sources of PDBPs include legumes, grains, nuts, fruits, vegetables, and so on [4,5]. Compared with animal sources, PDBPs have abundant sources, simple preparation and low cost, and have a good development trend [6,7].

It has been reported that PDBPs can obtain small bioactive peptides by enzymatic hydrolysis, microbial fermentation, chemical synthesis and other methods under specific conditions [8,9]. Among them, enzymatic hydrolysis and microbial fermentation are the most favorable biotechnological methods for releasing bioactive peptides [10]. PDBPs have specific biological functions that original proteins and amino acids do not possess. Their various biological activities, such as hypotensive, antioxidant, antibacterial, hypoglycemic, and anticancer properties, are constantly being explored [11]. The biological activity of PDBPs is related to their specific amino acid composition, sequence, quantity, position in the carbon chain and the spatial structure of the peptide chain [12]. Relevant studies have pointed out that PDBPs have the advantages of a low molecular weight and easy digestion and absorption in the intestine [13,14]. These characteristics determine their function to a certain extent, and, thus, have more favorable treatment effects for chronic diseases. For example, peptides extracted from rice bran inhibit the growth of bowel cancer cells (Caco-2), breast cancer cells (MCF-7), and liver cancer cells (HepG-2) [15]. In addition, peptides derived from grains, such as oats and barley, have a strong inhibitory effect on ACE [16]. Studies have found that PDBPs can be used to prevent and treat cancer, high blood pressure, and other chronic diseases. They have high edible and medicinal value and can meet the various needs of human health [17].

Notably, over the past decade, mentions of PDBPs have increased from 41 in 2014 to 205 in 2024 (Figure 1A). Figure 1B shows the co-occurrence of keywords associated with PDBPs. Among them, there are many keywords related to human chronic diseases, such as inflammation, blood pressure, cancer, and chronic diseases. It can be seen that researchers are increasingly exploring PDBPs, and the unique structural characteristics and functional mechanisms of PDBPs are increasingly attracting attention. Therefore, based on the latest research progress of PDBPs in recent years, this review focuses on the biological activity, structure–activity relationship, and application of PDBPs in chronic diseases. The preparation methods and limitations of PDBPs are also briefly introduced. It is expected to provide reference for further research and application development of PDBPs.

## 2. Methods

This narrative review searched the PubMed, Web of Science, Google Scholar, SpringerLink, and Science Direct databases using keywords and related terms. It used certain keywords, i.e., plants and bioactive peptides, and combined such terms with the following keywords: enzymolysis, fermentation, bioactivity, structure–activity relationship, and chronic diseases. The final search was conducted in June 2024. The literature search language was English, and references were selected according to their relevance. In this review, we first read the title of the high-value references in the literature, then searched by the title, and finally read the whole article. References that were not relevant to the review and replicated studies were excluded, and abstracts of the remaining articles were reviewed to ensure that they met the review’s inclusion criteria. Because this article is a narrative review, it is not necessary to document the literature searches on specific platforms [18].

## 3. Preparation of PDBPs

According to the different internal and external factors, such as environment, cost, and sample composition, different preparation methods are selected to obtain better PDBPs. At present, the common preparation methods of PDBPs include enzymatic hydrolysis and microbial fermentation. Therefore, we briefly introduce the advantages and disadvantages of the following two preparation methods (Table 1).

### 3.1. Enzymatic Hydrolysis

Enzymatic hydrolysis is one of the main methods of protein hydrolysis in the food industry. Its mild reaction conditions, high selectivity, and high safety have been proven to be an economical and effective strategy [19]. Its main method is to hydrolyze the amide bond sites of proteins by specific enzymes, so as to obtain different types of peptides, which is the most commonly used method for preparing plant bioactive peptides. Because the specificity of different enzymes is different, some low molecular peptides with biological activity can be obtained by using one or more proteolytic enzymes [20].

#### 3.1.1. Monoenzymatic Hydrolysis

In the study of monoenzyme hydrolysis, alcalase can specifically break peptide bonds. Therefore, compared with other enzymes, alcalase is often used for proteolysis from various plant sources. An et al. [21] proved that the optimal conditions for the preparation of wheat flour polypeptide by enzymolysis (alcalase) were as follows: the ratio of solid to liquid was 1:7, the amount of enzyme was 1.2%, the reaction time was 2 h, and the yield and purity of the peptide could reach 83% and 72%, respectively. In addition, Xiang and colleagues [22] obtained the best conditions for preparing seabuckthorn seed meal protein hydrolysate using alcalase: an initial pH = 10, a temperature of 50 °C, an enzyme dosage of 0.25 g/100 g protein, and a hydrolysis time of 8 h. Cottonseed protein hydrolysate obtained by alcalase showed antibacterial activity against four tested strains (*Staphylococcus aureus*, *Streptococcus*, *Escherichia coli*, and *Salmonella*) [23]. Golden melon seeds were hydrolyzed with alkaline protease to obtain peptide. Frac1 (molecular weight > 5 kDa), Frac2 (molecular weight 3~5 kDa), and Frac3 (molecular weight < 3 kDa) were obtained by ultrafiltration. Frac3 had the highest antioxidant capacity [24].

Other proteases are also used in the experiments of single enzymatic hydrolysis of proteins. Some scholars have studied and compared the enzymatic hydrolysis of a variety of different enzymes on the same plant. Tian et al. [25] used microwave-assisted enzymatic hydrolysis of wheat germ albumin to study the effects of alkaline protease, neutral protease, papain, and compound protease on its physical and chemical properties. The results showed that papain had the best proteolytic activity. The solubility increased and the apparent viscosity and foam stability decreased. A recent study found that hydrolyzed protein isolates from pigeon peas, lentils, and chickpeas could be extracted using alcalase and bromelain, and the study found that bromelain enhanced the water absorption and oil binding capacity of the three proteins. The potential of enzymatic hydrolysis in the production of functional components with antioxidant and anti-inflammatory properties in legume proteins was concluded [26]. In another study, Ren et al. analyzed the effects of five food-grade proteases, namely alkaline protease, neutral protease, trypsin, pepsin, and trypsin, on the antioxidant activity of rice bran protein hydrolysate. The results showed that trypsin hydrolysate had high iron chelation activity, DPPH and hydroxyl radical scavenging activity, and could improve the oxidative damage of Caco-2 cells induced by hydroxyl radicals [27]. In addition, in 2022, some scholars studied the five kinds of enzyme hydrolysis of mung bean protein enzymatic hydrolysis ability, including alcalase, neutrase, flavorzyme, papain, and protamex, and proved the highest alcalase hydrolysis degree. The physicochemical properties of five enzymatic hydrolysis products were evaluated by SDS-PAGE, particle size distribution analysis, FTIR, and ultraviolet visible and fluorescence spectrophotometry. It was also found that the products of protamex and papain had a higher foaming ability, emulsifying activity index, and emulsifying stability index [28]. The hydrolysates and fractions of *Ulva* sp. produced with food-grade papain have low renin and high ACE-1 inhibitory activities in vitro [29]. Similarly, Lu et al. [30] also used a variety of enzymatic hydrolysis comparison methods to explore the zinc-binding ability and in vitro gastrointestinal stability of Cucurbita pepo L after hydrolysis by different enzymes. The peptides hydrolyzed by papain had the largest average molecular weight, the smallest particle size, the highest hydrophobicity, and the largest zinc binding ability, and showed better stability.

#### 3.1.2. Complex Enzymatic Hydrolysis

Complex enzymatic hydrolysis is based on the action of a single enzyme and the combination of one or more proteases, according to the simultaneous hydrolysis of proteins or step-by-step hydrolysis of complex enzymatic hydrolysis. In 2024, an article was published on the dual enzyme hydrolysis of walnut pomace, which used aminopeptidase combined with alcalase hydrolysis to study the effect of different digestion methods on the flavor and antioxidant activity of walnut pomace protein products [31]. Chia seed by-products were hydrolyzed using alcalase and flavorzyme, and then hydrolyzed with alcalase and flavorzyme, respectively. It was found that chia seed protein obtained by using two enzymes (alcalase and flavorzyme) had the highest hydrolysis degree, and potential antioxidant, antihypertensive, and antithrombotic peptides could be obtained [32]. Other studies have shown that the combined enzymolysis of *Moringa oleifera* leaf protein (alcalase, neutrase, and flavorzyme) has stronger ACE inhibitory activity and DH than a single enzymolysis [33]. The optimal semi-solid enzymatic hydrolysis (SEH) parameters obtained for the preparation of adzuki bean peptides by SEH incorporating antioxidant activity, ACE inhibitory activity, and enzymatic efficiency included a ratio of alkaline protease–neutral protease = 2:1 (*w*/*w*) and enzyme addition of 3.75% (*w*/*w*) [34]. In addition, it was found that three soybean peptide groups prepared by different methods (alcalase + flavorzyme: hydrophobic amino acid-rich peptides; papain and flavorzyme: basic amino acid-rich peptides; and bromelain and flavorzyme: balanced amino acid compositions) had different sequence characteristics for their endonuclidenase cleavage sites [35]. Bioactive peptides with increased ACE inhibition and antioxidant activity have been produced by single and sequential hydrolysis of green lentil (*Lens culinaris*) seed. The results show that using a combination of enzymes in continuous hydrolysis can produce bioactive peptides with improved ACE inhibition and antioxidant activity, and with better antihypertensive and antioxidant activity [36]. It has been reported that using three different proteases (alkaline protease, papain, and trypsin), an encrypted and highly potent ACE-inhibitory peptide can be released from defatted moth bean powder from moth bean seeds [37].

#### 3.1.3. Microbial Fermentation

Microbial fermentation is a relatively old food biological technology [38]. It is not only an inexpensive way to produce bioactive peptides from protein substrates, but also to remove hyper-allergenic or anti-nutritional factors that may be present [39,40]. In the food industry, in order to obtain PDBPs with good physical and chemical properties, fermentation is also used to obtain stable and high-quality products [41]. The bioactive peptides obtained by the fermentation of red lentil protein isolate with various lactic acid bacteria and yeast strains have anti-free radical, angiotensin-converting enzyme inhibiting, and antifungal activities [42]. Serena et al. [43] used strains of lactobacillus to ferment a new pistachio beverage. According to the BIOPEP database, the resulting peptides were found to have potential biological activity, mainly related to antioxidant properties and ACE and DPP-IV inhibition. In addition, an in vitro experimental study found that the mixed fermentation of lactic acid bacteria and Aspergillus oryzae was used to prepare soybean meal. The optimal fermentation conditions were determined by response surface analysis. DPPH and ABTS free radical assays were used to determine the higher antioxidant activity of soybean peptides. At the same time, it was also found that it can effectively inhibit the hemolysis of red blood cells induced by AAPH, prevent the production of intracellular ROS, balance the antioxidant enzymes (SOD, CAT, and GSH-Px) system, and reduce the expression of MDA content [44]. *Brassica napus* is one of the top ten oil crops in the world, and after being pressed to make oil, the remainder is called the rapeseed meal. More recently, rapeseed meal has been fermented by a mixture of strains (*Bacillus subtilis, Pediococcus acidilactici*, and *Candida tropicalis*). Compared with the unfermented samples, the microbial diversity decreased significantly, indicating that mixed bacteria fermentation could inhibit the growth of mixed bacteria and increase the polypeptide content [45].

### 3.2. Separation and Purification

Plant protein hydrolysates and peptides can be separated by molecular weight, and low-molecular-weight protein hydrolysates often have better biological activity and can be used for further study. The protein hydrolysis product obtained after purification is purer, removing impurities and unhydrolyzed large proteins from the mixture [46]. The separation and purification of peptide mixtures have been well validated under laboratory conditions. However, the time required to go from enzymatic hydrolysis to separation and purification is long and the equipment is expensive. More exploration and process optimization are needed to improve the process. At present, the separation and purification methods of peptides used in the market include gel filtration chromatography, ion exchange chromatography, reversed-phase high-performance liquid chromatography (RP-HPLC), capillary electrophoresis (CE), ultrafiltration (UF), nano-liquid chromatography–mass spectrometry (nano LC-MS/MS) and other methods [47]. The separation and purification of peptides is usually first performed by ultrafiltration [48]. The technical principle is to remove macromolecular impurities by membrane separation technology, so as to obtain peptides of different molecular weights with different biological activities. However, due to the use of ultrafiltration technology, further separation and purification of the enzyme products are required. Therefore, it is difficult to obtain high purity peptides for further purification by chromatography [49]. For example, Dou et al. [50] isolated and purified peptides from Idesia polycarpa Maxim using ultrafiltration and dextran gel chromatography, combined with mass spectrometry to identify components with significant antioxidant activity, and screened and characterized peptides using various techniques, such as network pharmacology and molecular docking. In addition, Yasen et al. [51] isolated and purified *Cuminum cyminum* L. Seeds by 50% ethanol extraction, C18 reversed-phase column chromatography, and ion exchange chromatography for the first time, and obtained antimicrobial and antioxidant peptides. Then, the separated fractions were characterized by gel electrophoresis (SDS-PAGE) and high-pressure liquid chromatography, and the peptide components and molecular weight were determined by liquid chromatography and mass spectrometry. A novel antimicrobial peptide was prepared from cauliflower and digested and hydrolyzed with pepsin by ultrafiltration, molecular sieve chromatography, ion exchange chromatography, and mass spectrometry. The best active hydrolysates were further purified from the Changii Radix hydrolysates by ultrafiltration, size exclusion chromatography, and semi-preparative high-performance liquid chromatography [52].The peptides in the most active portion were then identified and screened for biological activity by nanoscale LC-MS/MS. The enzymolysis products from *Limnospira maxima* protein were isolated and purified with a 10 kDa range of peptides using ultra-membrane filtration, SDS-PAGE, and TLC. The resulting products showed antibacterial properties against *Escherichia coli* and *Staphylococcus aureus* [53]. In a study, the ACE inhibitory peptide HPVTGL was identified, which is derived from the protein hydrolysis product of rape (*Brassica napus* L.) bee pollen. It was obtained mainly by ultrafiltration separation and purification using preparative high-performance liquid chromatography [54].

In conclusion, the separation and purification of PDBPs is a complex process, which requires comprehensive consideration of the preparation method, separation and purification technology, and identification technology of bioactive peptides. A single separation and purification method may be limited by equipment. A combination of techniques is often used to achieve better separation of peptide mixtures, resulting in high-purity peptides as well as higher bioactivity peptides. With further advances in technology, we expect to obtain bioactive peptides from plants more efficiently, contributing to human health.

## 4. Bioactivities of PDBPs

PDBPs are hydrolyzed products obtained from plants. PDBPs have powerful functions, but their structures can be as simple as two amino acids or as complex as twenty amino acids [55]. The advantages of PDBPs are mainly focused on the two aspects of being more easily absorbed by the human body and possessing physiological functions far beyond those of amino acids. As shown in Figure 2, the biological activities of PDBPs mainly include antioxidant, anticancer, antibacterial, antihypertensive, hypoglycemic, immunomodulation, and the regulation of intestinal flora and so on.

### 4.1. Antioxidant Activity

Redox imbalance produces reactive oxygen species. However, excessive production of reactive oxygen species or insufficient scavenging capacity may lead to accumulation of oxidative stress, which in turn triggers inflammation, cellular damage, tissue senescence, and decreased organ function [56]. Studies have shown that natural antioxidants maintain cell health by neutralizing harmful free radicals. Furthermore, PDBPs are a good source of natural antioxidants, which can neutralize excess free radicals and stabilize their electrons, thus protecting cells from oxidative damage. They are also beneficial for preventing or treating oxidative stress-related sub-health or disease states [57]. The Keap1/Nrf2 signaling pathway has attracted much attention in studies of the antioxidant mechanisms of PDBPs (Figure 3). Currently, PDBPs are of great interest because of their high safety, good absorption, and wide array of sources. As shown in Table 2, bioactive protein hydrolysates or peptides have been obtained from different plant sources, such as seeds, nuts, leaves, and their plant by-products.

### 4.2. Antibacterial Activity

With the proliferation of antibiotics and chemical pesticide use, there is a growing problem of resistance in flora as well as in antibiotic residues. This has become one of the factors jeopardizing public health. Therefore, there is an urgent need for alternatives to antibiotics to reduce antibiotic resistance [73]. In recent years, antimicrobial peptides have emerged as promising drug candidates against drug-resistant pathogens. Researchers have been working to find antimicrobial peptides of natural origin. Typically, antimicrobial peptides are categorized according to different criteria including their source, structure, activity, and amino acid composition (Figure 4). A variety of antimicrobial peptides have been isolated from microorganisms, insects, amphibians, plants, and mammals [74]. Among them, plants are a promising source of antimicrobial peptides. Compared with antibiotics, antimicrobial peptides of plant origin have advantages, such as low toxicity and high efficiency, as well as a broad spectrum and structural diversity. They are present in almost all areas of life as part of the innate immune system. They can fight viruses, bacteria, fungi, and even cancer cells, and are the first line of defense against invading pathogens [75]. It has been shown that bioactive peptides obtained from hydrolyzed adzuki bean and mung bean exhibited antibacterial activity against *Salmonella typhimurium* and *Staphylococcus aureus*, respectively [76]. Kong et al. [77] hydrolyzed cottonseed protein and identified three novel bacteriostatic peptides, HHRRFSLY, KFMPT, and RRLFSDY, by isolation and purification. These three peptides can achieve bacteriostatic effect by disrupting the cell membrane of *Escherichia coli*. In addition, Hu et al. [78] prepared bacteriostatic peptides from fermented walnut meal and identified four bacteriostatic peptides, FGGDSTHP, ALGGGY, YVVPW, and PLLRW. Antimicrobial peptide was obtained from sorghum spent grain hydrolysate. It can counteract the activities of two Gram-negative and three Gram-positive bacteria via the minimum inhibitory concentration method [79]. *Moringa oleifera* seed protein hydrolysates were identified as a novel antimicrobial peptide named MOp2 (HVLDTPLL), which can cause irreversible membrane damage to *Staphylococcus aureus* by increasing membrane permeability, leading to the release of the intracellular nucleotide pool. In addition, molecular docking showed that MOp2 can inhibit *Staphylococcus aureu* growth by interacting with dihydrofolate reductase and DNA lyase through hydrogen bonding and hydrophobic interactions [80].

### 4.3. Anticancer Activity

Cancer is the second leading cause of death worldwide after cardiovascular disease. However, some anticancer drugs have significant side effects and are prone to damaging normal cells [81]. Therefore, the development and utilization of natural anticancer peptides is an inevitable trend. Currently, the potential of bioactive peptides as anticancer agents of natural origin has been widely reported based on in vitro and in vivo tests. They have been demonstrated to have various anticancer effects on mature cancer cell lines, including the inhibition of cell migration, the inhibition of angiogenesis, antioxidant properties, the inhibition of cell proliferation, the induction of apoptosis, and cytotoxicity [82]. Meanwhile, bioactive peptides obtained from plant sources have received increasing attention for their potential as complementary therapies and have been promoted as promising therapeutic treatments for a variety of human diseases, especially due to their bioactivities with anticancer potential. Several studies have shown that bioactive peptides derived from a variety of plant proteins, such as those obtained from legumes, algae, garlic and rice berry, have antiproliferative and toxic effects on cancer cells, favoring cell cycle arrest and apoptosis. Mung bean protein hydrolysate, obtained by the hydrolysis of papain, has been shown to inhibit the proliferation of mouse tumor cells without the side effects of chemical anticancer drugs, and mung bean peptide promotes apoptosis in HepG2 cells and blocks the cell cycle at a lower S phase [83]. The hydrolyzed product of black soy protein was isolated and purified by ultrafiltration and chromatography. The amino acid sequence was identified as Leu/Ile-Val-Pro-Lys, which showed high cytotoxic potential against HepG2, MCF-7, and Hela cells. Molecular docking studies showed that the purified peptide efficiently bound to four apoptosis-related key proteins (XIAP, caspase-3, caspase-7, and Bcl-2) through hydrophobic interaction and hydrogen bonding [84]. It has been reported that some components of Mucuna pruriens beans peptides show high activity to protect DNA damage. 5–10 kDa exhibited significant cytotoxic activity against HepG2 and QGY-7703, and the related gene protective effects were thought to be significantly different [85]. Chickpea peptide induced S phase and G2 phase cell cycle arrest in a dose-dependent manner. DNA breakage and apoptosis were induced by downregulating Bcl-2 expression, upregulating Bax expression, and promoting caspase-3 activation [86]. Similarly, bioactive substances found in algae have effective anticancer properties. This promising and prolific source works primarily by causing apoptosis and by preventing cell division through disrupted signaling pathways. Among the identified bioactive compounds, peptides, such as VECYGPNRPQF from Chlorella, have shown significant anti-proliferative effects, especially on the gastric cancer cell line AGS, underlining their potential for further development in cancer therapy [87]. In addition, three phycocyanin-derived bioactive peptides with predicted anticancer ability were identified to significantly inhibit the growth and migration of A549, H1299, and LTEP-a2 cells. They also inhibited Akt pathway activity in NSCLC cells [88]. Another in vivo study found that LCP-3 [cyclo-(Trp-Leu-His-Val)] isolated from kelp inhibited the growth of colon cancer and induced apoptosis in cancer cells of loaded mice using an anticancer activity tracking assay [89]. Recently, scholars have identified the anticancer activity of peptides isolated from garlic against leukemia cell lines, and have also discovered a novel anticancer peptide, VKLRSLLCS. It has been shown to significantly inhibit the proliferation of MOLT-4 and K562 leukemia cell lines, and has apoptosis-inducing properties on leukemia cell lines through the anti-apoptotic Bcl-2 protein family [90]. The hydrolyzed rice bran extract had inhibitory effect on colon cancer cell line but had little effect on normal cells. Riceberry rice bran protein hydrolyzed fractions can induce apoptosis of metastatic cancer cells and senescence of non-metastatic cancer cells [91].

### 4.4. Antihypertensive Activity

Hypertension is an important global health problem affecting about one-third of the world’s adult population, and it is a high risk factor for cardiovascular diseases, such as end-stage renal disease, stroke, atherosclerosis, and myocardial infarction [92]. As shown in Figure 5, the two major systems regulating blood pressure are the renin-angiotensin system (RAS) and the kallikrein-kinin system (KKS). ACE plays an important role in the regulation of blood pressure, mainly acting on angiotensin I. It can produce angiotensin II through the RAS, which in turn inactivates bradykinin in the KKS, ultimately leading to an increase in blood pressure. Therefore, inhibition of ACE activity can play a role in lowering blood pressure [93]. At present, more studies have confirmed that specific peptides obtained by enzymatic digestion have strong ACE inhibitory activity. This has been demonstrated by the Red Alga *Acrochaetium* sp. screening of bioactive peptides. The protein hydrolysate was isolated by chromatography, and VGGSDLQAL (VL-9) was identified. The peptide VL-9 shows the ACE inhibitory activity with IC50 value 433.1 ± 1.08 µM [94]. Duan et al. [95] isolated and characterized the novel ACE inhibitory peptides FQW, FRW, and CPF from rapeseed. They showed strong ACE inhibitory activity in vitro with IC50 values of 46.84 μmol/mL, 46.30 μmol/mL, and 131.35 μmol/mL, respectively. Moreover, these peptides can interact with ACE active sites through hydrogen bonding and hydrophobic interaction. Corn gluten meal protein hydrolysate was found to not only inhibit ACE and renin activity, but also to regulate the biosynthesis and metabolism of fatty acids, sex hormones, and aldosterone through a rat model of spontaneous hypertension, thus contributing to a reduction in hypertension. In addition, Zou et al. [55] also found that wheat bran protein hydrolysates had a significant inhibitory effect on renin and ACE in rats under the condition of constructing a rat model of spontaneous hypertension [96]. Some researchers have identified some promising plant-derived raw materials for the preparation of antihypertensive peptides. SNHANQLDFHP and PVQVLASAYR were identified from pumpkin seed meal hydrolysate. The two peptides performed a protective function on EA. hy926 cells by decreasing the secretion of endothelin-1, increasing the release of nitric oxide, and regulating the ACE 2 activity [97]. *Azolla pinnata* fern protein is deeply hydrolyzed (30%) by alkaline protease to produce hydrolysates with antihypertensive (ACE inhibitory) activity [98]. Peptides extracted from water lentils [99], oil palm kernel [100], green basil leaves [101], and quinoa [102] have been found to have antihypertensive activity.

### 4.5. Hypoglycemic Activity

Diabetes mellitus is a serious and complex chronic metabolic disease characterized by elevated blood glucose due to insulin resistance or inadequate insulin secretion [103]. Recent processing technologies and human nutrition have identified plant proteins as an important source of food-derived bioactive compounds. These bioactive peptides are potential active ingredients. Among them, bioactive peptides with hypoglycemic activity can alleviate the hyperglycemic state and achieve blood glucose lowering without the help of drugs [104]. As shown in Figure 6, α-amylase, α-glucosidase, and so on are key enzymes that regulate blood glucose. Inhibiting the activity of these enzymes is considered an effective strategy for controlling diabetes. It has been shown that the peptides TGGR, SPVI, FY, and FR obtained from hemp (*Cannabis sativa* L.) protein by identification, molecular docking, and virtual screening exhibited good α-glucosidase inhibitory activities, respectively. Animal experiments showed that these peptides could regulate blood glucose and blood lipids in hyperglycemic rats [105]. Multi-enzyme hydrolyzed *Amygdalus communis* L. purified by ultrafiltration showed the best inhibitory activity of active peptide B4 against α-amylase and α-glucosidase [106]. TGPs-75 obtained from *Torreya grandis* meal peptides significantly reduced blood glucose concentrations. According to transcriptome analysis, 382 genes were significantly differentially expressed after TGPs-75 pretreatment. The main functions of these genes were related to gluconeogenesis and insulin resistance. This study suggests the possibility of using peptides from camellia seeds in camellia seed cake as hypoglycemic compounds for the prevention and treatment of diabetes [107]. Rapeseed-derived ELHQEEPL showed significant dipeptidyl peptidase-IV (DPP-IV) inhibitory activity [108]. Currently, there are also legume peptides and hydrolysates that have shown significant hypoglycemic effects by various methods [109]. For example, soybean peptides produced by alkaline protease hydrolysis exhibited the highest hypoglycemic activity by Xu et al. By analyzing their molecular weight distribution and amino acid composition, a positive correlation between the aromatic and hydrophobic amino acid content of the alkaline protein hydrolysate and the hypoglycemic activity was found [110].

### 4.6. Immunoregulatory Activity

The immune system is an autonomous defense system that prevents and controls various infectious diseases. Diseases caused by an imbalance in immunomodulation affect the body’s immune response [111]. However, many immunomodulatory drugs are not suitable for long-term or prophylactic use. Immunomodulatory peptides have been reported from various plant sources, such as soybean, lotus seed, rice, hemp, corn, pea, and others. This offers potential options for the treatment of immune-related diseases and the development of immunomodulatory therapies. It has been noted that soy peptides have multiple immunomodulatory activities on both innate and adaptive immune responses [112]. Soy peptides were prepared by hydrolyzing soy protein by alkaline protease. Furthermore, sequences of 51 peptides were identified by UPLC-MS/MS, of which 46 peptides were designated as immunomodulatory peptides. They could promote macrophage proliferation, increased cytosolic activity, and NO levels [113]. In addition, the low-molecular-weight peptide (<3 kDa) from lotus seeds had the greatest effect on the phagocytosis of RAW264.7 macrophages relative to other molecular weight peptides (>10 kDa, 5–10 kDa, 3–5 kDa). It also significantly elevated the amounts of NO, IL-6, and TNF-α secreted by peritoneal macrophages in immunosuppressed mice [114]. Immunomodulatory peptide (YGIYPR) was prepared from rice protein hydrolysate. It enhanced the proliferation of macrophage RAW 264.7 cells in the range of 12.5–100 μg/mL, and good proliferation was achieved even at the smallest doses [115]. The immunomodulatory properties of hemp protein hydrolysates (HPH) may have beneficial effects on intestinal epithelial levels [116]. Similarly, maca belongs to a group of protein-rich edible plants with immunomodulatory activity. Maca protein hydrolysate enhances phagocytosis and the secretion of NO, TNF-α, and IL-6 by RAW 264.7 cells [117]. There are also studies that have identified immunomodulatory peptides from corn proteins using human macrophage-like U937 cells [118]. In addition, pea protein hydrolysates could significantly increase the immunomodulatory activity of the macrophages by elevation of phagocytic activity and promoting the production of nitric oxide and pro-inflammatory cytokines (TNF-α and IL-6) [119].

### 4.7. Regulation of Gut Flora

The various microorganisms that live in the gut form a complex, dynamically balanced intestinal flora. This intestinal barrier is an essential defense measure in the gut, providing strong protection against harmful microorganisms both internally and externally [120]. In recent years, it has been widely recognized that gut microorganisms play an important role in a variety of human diseases and are considered to be the “second genome” of the human body [121]. Studies have shown that bioactive peptides can prevent the proliferation of external bacteria and viruses, thereby maintaining the balance and stability of the gut microbiota [122]. Currently, bioactive peptides can be obtained from plants to regulate gut flora (Figure 7). Walnut protein-derived peptide LPF has protective and restorative effects on DSS-induced colitis in mice. After LPF was administered, the composition of intestinal flora in mice was continuously improved during the convalescence period of colitis. The relative abundance of beneficial bacteria increased, while the abundance of potentially harmful bacteria decreased [123]. Similarly, walnut peptide (WP) can increase the level of bacteroides and decrease the level of firmicutes in the intestinal flora of obese mice. At the same time, the abundance of the genus *Adlercreutzia* spp., which is positively associated with BMI and inflammation, decreased significantly, and *Lactobacillus* and *Butyricimonas*, which have been shown to prevent obesity and regulate intestinal microecological balance, increased significantly across the entire bacterial genus group [124]. Zhang et al. [125] analyzed the regulation and influence of cardamom selenium-containing peptides on intestinal microflora. The abundance of *Akermania*, *Firmicutes*, and *Bacteroidetes* was higher in mice treated with cardamom antibiotics containing selenium peptide, while the abundance of proteobacteria was lower. In another study, rice glutelin peptides helped to increase *Enterococcus* and *Bacteroides* in the gut, while *Streptococcus*, *Lactobacillus*, *Faecalibacterium*, *Parasutterella*, and *Turicibacter* were decreased [126]. Ginseng peptide (GP) can improve intestinal flora disorder in T2DM mice by regulating ginseng peptide abundance. *Ruminococcus* and *Bifidobacterium* are the main hypoglycemic bacteria affected by GP-H effect [127]. In addition, Li et al. [128] significantly improved the abundance and homogeneity of intestinal flora after feeding with soybean-derived peptides (SPep) for 35 days. SPep significantly promoted the growth of *Lactobacillus* and *Phascolarctobacterium*.

### 4.8. Other Biological Activities

In addition to the biological activities mentioned above, PDPBs have been found to possess anti-inflammatory activity and anti-fatigue activity, to improve neurodegenerative diseases, have anti-osteoporosis activity, have cholesterol-lowering activities, and so on. For example, hazelnut protein-derived peptide LDAPGHR has been shown to possess not only immunomodulatory activity, but also effective anti-inflammatory activity by inhibiting the production and mRNA expression of IL-1β, IL-6, and TNF-α [129]. In the weight-bearing swimming test, mice fed pea peptides were able to swim longer than the control mice [130]. LPF, GVYY, and APTLW isolated from walnut protein hydrolysate by chromatography have been reported to ameliorate LPS-induced memory deficits by normalizing inflammatory responses and oxidative stress in the brain [131]. In another in vivo experiment, wheat germ peptide ADWGGPLPH can effectively reduce the level of oxidative stress and improve microstructure and bone density in aged osteoporotic rats. In addition, ADWGGPLPH can enhance the proliferation and differentiation of osteoblasts and inhibit the differentiation of osteoclasts by regulating the OPG/RANKL/RANK/TRAF6 pathway [132]. At the same time, Yuan et al. [133] found that flaxseed peptide (FP5) with a molecular weight of less than 1 kDa can reduce cholesterol absorption and synthesis. The serum and liver cholesterol levels were significantly reduced after FP5 supplementation in rats with high cholesterol and high fat. FP5 improves the mechanism of liver cholesterol metabolism, inhibits de novo cholesterol synthesis, promotes bile acid synthesis and excretion, and inhibits bile acid reabsorption.

## 5. Structure–Activity Relationship of PDBPs

The complex biological structure and diverse functions of peptides are mainly determined by the number, composition, and arrangement of amino acids and the spatial structure of peptide chains. As shown in Table 3, the structure–activity relationship of PDBPs extracted and isolated from plants were identified.

### 5.1. Molecular Weight

It is well known that extensive hydrolysis produces small peptides (1–3 kDa) that are more likely to be bioavailable and biologically active. Chen et al. [148] explored the effect of molecular weight on the antioxidant capacity of rice protein hydrolysate. Corn gluten meal hydrolysate was obtained by hydrolysis using a combination of alcalase, flavorzyme, and protamex, and the hydrolysate was analyzed for antioxidant activity after further fractionation. Corn peptides with a molecular weight less than 1 kDa showed excellent antioxidant activity [149]. In particular, peptides with molecular weights less than 3 kDa showed better ACE inhibition compared to larger peptides. A low-molecular-weight (LMW) fraction (<3 kDa) was extracted from *Olea europaea* (cv. Farga). The LMW grade has a strong ACE inhibitory activity [150]. In addition, low-molecular-weight peptides were prepared from soybean dregs using high-pressure homogenization-assisted dual enzymes, the structure of which resulted in changes only in the hydrogen bonding between peptide chains. It promoted cellular phagocytosis, NO levels, and the release of cytokines IL-6, IFN-γ, and TNF-α [151]. Another study found that most of the peptide fragments of low-molecular-weight peptides extracted from red macroalgae possessed different levels of antioxidant, antimicrobial, anti-ACE, and anti-DPP-IV inhibitory activities [152]. A preparation of oat peptides using in vitro simulated digestion produced low-molecular-weight antimicrobial peptides that ameliorated colitis in rats [153].

### 5.2. Amino Acid Composition and Sequence

In terms of amino acid composition, the biological activity of PDBPs is related to the proportion of acidic and basic amino acids, hydrophobic amino acids, and aromatic amino acids that they contain [154]. Take antioxidant peptides as an example: a recent study has shown that chemical groups on amino acid residues are related to the antioxidant activity of peptides. Phenolic hydroxyl groups in Tyr, indole groups in Trp, and imidazole groups in His of peptides may increase their interaction with free radicals [155]. In addition, aromatic amino acid residues (Tyr, Trp, and Phe) have been reported to play a key role in the antioxidant activity of peptides [156]. Phenolic hydroxyl groups in tyrosine and nitrogen-hydrogen bonds in the indole ring of tryptophan of corn gluten meal hydrolysate are key sites for antioxidant activity. These findings may explain the effective chemical antioxidant activity [157]. Cottonseed is rich in acidic and basic amino acids, as well as aromatic and hydrophobic amino acids, which may contribute to its antioxidant capacity [158].

In amino acid sequences, peptides with similar chain lengths exhibit different functional activities according to their *C*-terminal and *N*-terminal amino acid sequences. Molecular docking studies of the three peptides of quinoa protein hydrolyzate revealed that the presence of specific amino acids in the peptide sequence (Pro, Phe, and Arg at the *C*-terminal, and Asn at the *N*-terminal) may contribute to the interaction between ACE and peptides [159]. Another study found that the *C*-terminal extension of SbGPRP1 from sorghum bicolor could act as an antimicrobial peptide by targeting bacterial outer membrane proteins [160].

### 5.3. Secondary Structure

The secondary structure usually refers to the spatial direction of the protein peptide chain along the backbone of the main chain, the regular cyclic arrangement, or a section of the peptide chain of the local spatial structure. That is, the relative spatial coiling and folding positions of the main chain of the peptide chain or the backbone atoms of a section of the peptide chain, which does not involve the conformation of the side chain of amino acid residues [161]. The common secondary structures are α-helix, β-fold, β-corner, and random curling. The stepwise enzymatic hydrolysis of coix seed prolamins promoted the destruction of the secondary structure of the hydrolyzed products, enhanced the β-angle structure, and increased the DPP-IV inhibitory activity [162]. Dual enzyme digestion products of walnut dreg showed an increased β-fold content and an altered bitter flavor intensity [31]. Habinshuti et al. have shown that the compactness of the peptide structure affects its antioxidant properties [163]. Similarly, Tia et al. found a decrease in β-folding from 33.74% to 15.64% and an increase in β-turning angle from 30.93% to 47.36% after the hydrolysis of malt albumin by ultrasound-assisted papain. At the same time, DPPH could reach 82.29% [164]. In addition, Liang et al. showed that changes in the secondary structure of active peptides directly affect the antioxidant activity of peptides [165]. Hydrolyzed soybean meal hydrolysates showed an increase in the number of β-turns and a decrease in α-helices and exhibited high antioxidant and ACE inhibitory activities [166]. Mulberry leaf protein hydrolysate, which consists mainly of disordered convolutions and β-folds, has antioxidant activity [167].

## 6. Application of PDBPs in Chronic Diseases

The full name of chronic disease is chronic non-communicable disease, and does not specifically refers to a disease, but rather refers to a class of diseases which have a long course and complex causes, and which cause health damage and serious social harm [168]. There is a lack of evidence of an exact infectious biological cause for such diseases; the etiology is complex, and some diseases have not been fully identified. At present, the complexity of chronic diseases poses a great challenge to our understanding of such diseases [169]. Cardiovascular and cerebrovascular diseases, diabetes, cancer, mental disorders, and other diseases are increasing and affecting people’s quality of life [170,171,172]. Many of these diseases are linked to lifestyle, environment, diet, or genetic factors. As a result, the scientific community’s study for new drugs and functional foods that can combat or prevent chronic diseases has increased over the past few decades [173]. Some natural resources are particularly attractive when environmental, cost, and renewable factors are taken into account [174].

In the context of the great popularity of plant products, PDBPs and their derived peptides stand out as promising bioactive ingredients. PDBPs can be produced from plant proteins by a series of methods. In addition, compared with animal-derived bioactive peptides, PDBPs also have advantages in human health [175]. More and more attention has been paid to their advantages, such as their non-toxic nature, low cost, wide availability, and diverse functions. Studies have shown that PDBPs can fight cardiovascular disease, type 2 diabetes, and some cancers, among others [176]. Additionally, the potential for using PDBPs to develop functional foods and drugs is huge. As far as we know, many antihypertensive drugs can cause side effects, such as dizziness [177]. As a result, the search for antihypertensive biopeptides from plants has increased. Wang et al. [178] evaluated the safety and antihypertensive activity of rapeseed peptides. The potential synergies with captopril were also discussed. In in vitro experiments, it was found that their synergistic effect further increased the serum levels of NO and endothelial nitric oxide synthase in rats. In another study, Lai et al. [179] optimized the extraction of green tea residue by alkaline extraction combined with enzymatic hydrolysis. Under optimized hydrolysis conditions, the ACE inhibition rate of the product was found to be 77.00%, and the molecular weight distribution of the polypeptide in the hydrolyzed product was widely in the range of 45.0–1.2 kDa. At the same time, in vivo experimentation also confirmed the good blood pressure lowering activity of this peptide. ACE inhibitory peptides were purified and identified from cashews, among which FETISFK showed the highest ACE inhibitory rate (91.04 ± 0.31%). It also reduced the expression of angiotensin II and angiotensin II type 1 receptors [180]. Type 2 diabetes is the dominant form of diabetes, accounting for more than 90% of the patient population. For clinical treatment, α-glucosidase inhibitors are a type of T2DM alternative therapy to regulate postprandial blood glucose levels [181]. At present, α-glucosidase inhibitory peptides have been isolated and identified from many natural sources. Three novel potential bioactive peptides (RWPFFAFM, AAGRLPGY, and VVRDFHNA) were screened from mulberry leaf proteins to inhibit α-glucosidase [182]. In addition, the α-glucosidase inhibitory activity of WGPs was identified with an IC 50 value of 6.87 mg/mL. LDLQR, AGGFR, and LDNFR were synthesized for further identification. The results of molecular docking and amino acid composition analyses showed that the high content of *C*-terminal Arg residues in the peptide may be the essential reason for its inhibition of α-glucosidase activity [183]. Recently, a highly active α-glucosidase inhibitory peptide with the amino acid sequence KETTTIVR was identified from Moringa oleifera seed protein hydrolysates, which was found to be a amphiphilic peptide with a β-corner structure [184]. PDBPs also contain anticancer peptides that help to improve and prevent the occurrence of a range of chronic cancers, more in line with the new generation of consumers’ pursuit of a healthy diet [185]. For example, cyclic peptides isolated from marine cyanobacteria, such as *Urumamide*, exhibited low proliferative inhibitory activity on human cancer cells [186]. Rapeseed peptide inhibited cell proliferation by regulating the P53 signaling pathway, inducing G0/G1 phase arrest, and the mitochondrial apoptosis pathway [187]. Recently, a peptide has been isolated that regulates the proliferation and invasion of non-small cell lung cancer cells. This polypeptide is derived from Bryopsis plumosa. Kim et al. obtained anticancer peptides through the experimental steps of peptide design, synthesis, and purification. In vivo experiments, peptides can effectively inhibit the metastasis of tumor xenografts in zebrafish embryos [188]. The most studied peptide from legumes is lunasin. Over the past decade, it has been identified that it exerts its anticancer action through different pathways, including regulating cell growth and apoptosis, epigenetic mechanisms, and blocking the transformation of normal cells. Diego Luna-Vital also summarized that different legumes have been used to produce peptides with anti-gastrointestinal cancer potential, especially soybeans, mung beans, lentils, chickpeas, and common beans. Most studies have focused on evaluating the anti-proliferation effects of plant-bioderived peptides on gastrointestinal cancer cells in vitro [189]. In addition, the isolation and relative purification of bioactive peptides from *Achillea eriophora* showed that the peptide mixture inhibited the growth of MCF-7 cancer cell lines, and also showed DPPH radical scavenging activity and the inhibition of copper ion reduction [190].

Currently, many PDBP in vitro cell experiments have shown a variety of activities that have been fully confirmed. However, studies in vitro have been very limited. Their safety needs to be further explained. Therefore, it is necessary to study PDBPs in vivo.

## 7. Limitations

The research related to PDBPs is gradually increasing, but PDBPs are not used in clinical and food and pharmaceutical applications on a large scale. There are many challenges in the development of PDBPs, mainly due to the following reasons. The first is the difficulties involved in in the preparation of PDBPs. The PDBPs obtained by different preparation methods, either enzymatic or fermentation, are structurally different and the peptides obtained cannot be determined. Considering the cost, it is impossible to select the conditions of enzymatic digestion or fermentation in a targeted manner. Secondly, various factors, such as time, temperature, and pH, can be a stumbling block to the preparation of PDBPs. In addition, how to ensure the yield of PDBPs is also an important indicator that can be successfully applied to the clinic. At present, PDBPs mostly stay in in vitro experiments. In the reviewing related literature, this review found that there are relatively few in vivo studies on PDBPs compared to in vitro studies. However, the development of chronic diseases in humans is very complex, and further animal studies and randomized clinical trials are needed to further explore the pathogenesis and therapeutic mechanism of these diseases. In the future, these limitations should be deeply explored and overcome in order to realize a wider application of PDBPs in the prevention and treatment of chronic diseases.

## 8. Conclusions

PDBPs has attracted attention in many fields because of their diverse biological activities, great potential, and wide application prospects. By promoting PDBPs, we can reduce our dependence on animals, reduce greenhouse gas emissions and water consumption, and protect the ecological environment. At the same time, the promotion of PDBPs can bring about healthier and more environmentally friendly food choices for humans. The current preparation methods (enzymatic hydrolysis and microbial fermentation) have been successfully applied to the extraction of active peptides, but still need further extraction and separation. In addition, the signal pathway of how PDBPs exert their biological activity through the body metabolism, as well as their potential targets and functional mechanisms, still need to be further explored. Therefore, future research should focus more on the prevention and treatment of human chronic diseases. This is of great significance for understanding the targeted action mechanism of peptides, optimizing their biological activity, and accurately designing functional foods or drugs.

## Figures and Tables

**Figure 1 nutrients-16-03277-f001:**
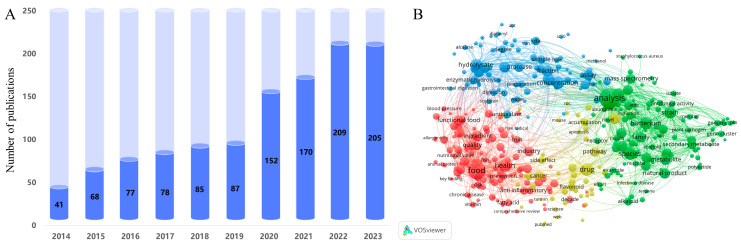
(**A**) Number of PDBPs-related publications in the Web of Science core collection in the past decade. (**B**) Keyword co-occurrence map based on bibliographic data created through VOSviewer 1.6.20 software (PDBPs, 2014–2023).

**Figure 2 nutrients-16-03277-f002:**
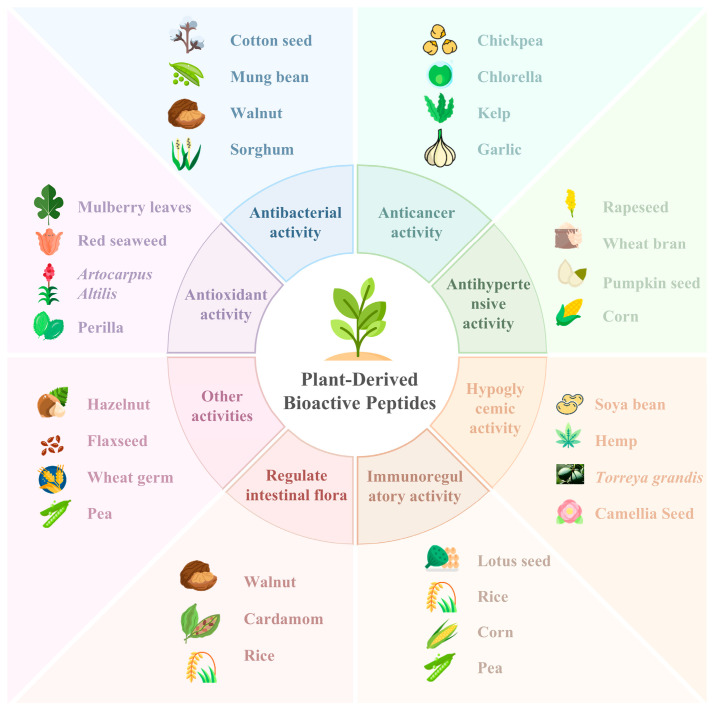
Multiple bioactivities of PDBPs.

**Figure 3 nutrients-16-03277-f003:**
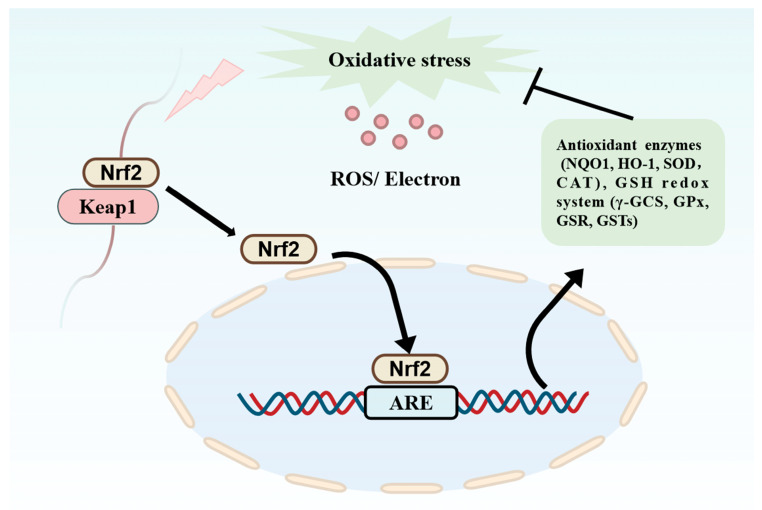
Keap1/Nrf2 signaling pathway.

**Figure 4 nutrients-16-03277-f004:**
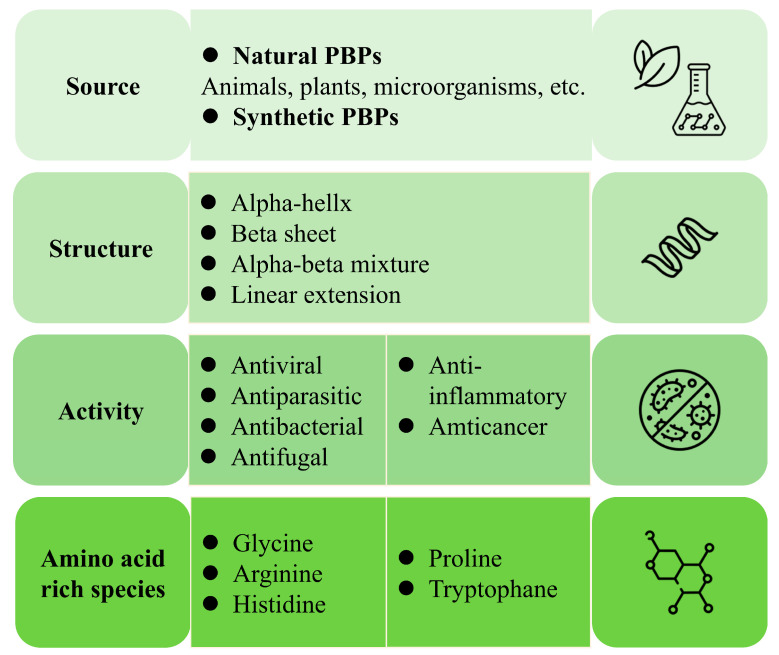
Antimicrobial peptides are classified according to different criteria.

**Figure 5 nutrients-16-03277-f005:**
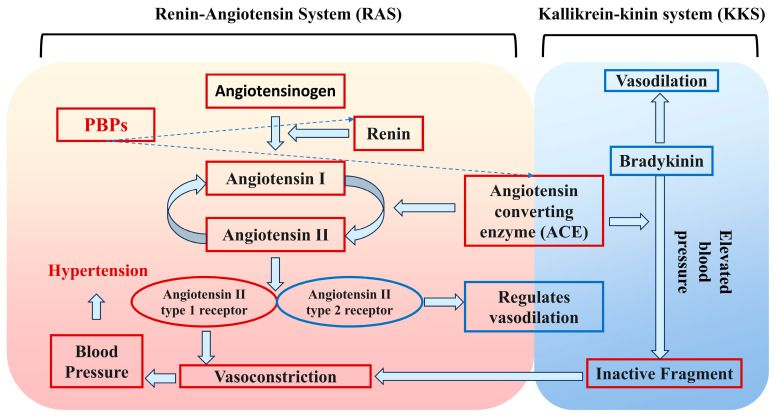
Two systems that regulate blood pressure.

**Figure 6 nutrients-16-03277-f006:**
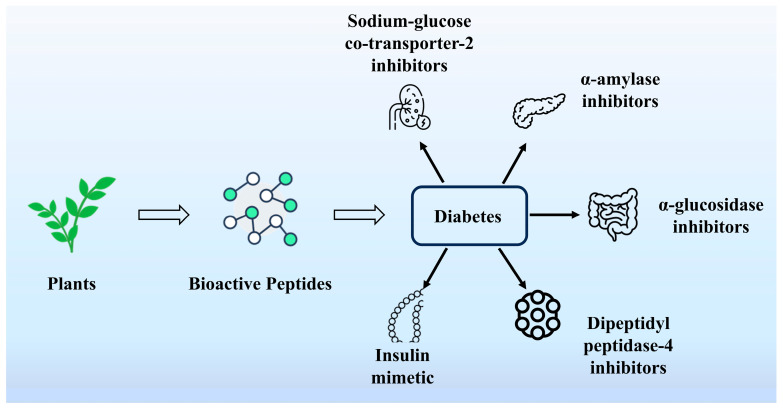
Several key enzymes that regulate blood sugar.

**Figure 7 nutrients-16-03277-f007:**
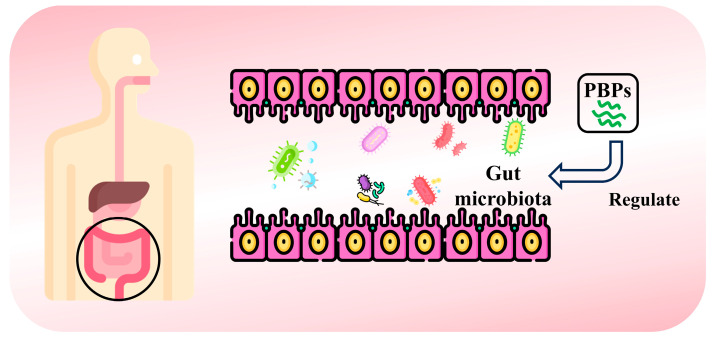
Schematic illustration of obtaining bioactive peptides from plants to regulate gut microbiota.

**Table 1 nutrients-16-03277-t001:** Advantages and disadvantages of two preparation methods.

Preparation Method	Advantages	Disadvantages
Enzymatic hydrolysis	No toxic substances are produced Non-destructive amino acid High catalytic efficiency High specificity	High cost
Microbial fermentation	High efficiency Low cost	Difficult to control Susceptible to microbial contamination Difficult separation and purification

**Table 2 nutrients-16-03277-t002:** Antioxidant activity and effects of PDBPs.

Plant Origins	Sequence/Name	Major Findings	Reference
Pea Protein	YLVN EEHLCFR TFY3	Through computer simulation of molecular docking, it was found that the anti-oxidation mechanism may be highly related to the activation of the Keap1-Nrf2 pathway by occupying the Keap1-Nrf2 binding site.	[58]
Mulberry leaves	EGDAGAQGPPGPAGPAGER RPGPSPGVGAPGK	↓ ROS ↑ SOD and CAT Both peptides are found to exert protective effects against H_2_O_2_-induced chromatin damage and cell apoptosis.	[59]
Red seaweed (*Palmaria palmata*)	SLLYSDITRPGGNMYTTR (SR18)	The SR18 peptide was found to have extremely high ROS scavenging activity and high ferric reducing ability and may be partially driven by the amino acids Tyr, Asn, and Met.	[60]
*Artocarpus altilis* (Parkinson) Fosberg	AAPPH	The different enzymatic hydrolysis products of the hydrolysis had a higher Fe^2+^-chelating capacity, DPPH scavenging capacity, and hydrogen peroxide scavenging capacity.	[61]
Rice bran	AFDEGPWPK	AFDEGPWPK has the ability to scavenge ORAC and DPPH free radicals, and it can enter the binding pocket of the Kelch structural domain and activate the Keap1/Nrf2/HO-1 pathway.	[27]
Pea	Pea-derived peptides (PPs)	↓ SOD, GR, GSH, GSSG, and ROS	[62]
Sacha inchi (*Plukenetia volubilis* L.)	SIH20B AAGALKKFL LGVKFKGGL	Antioxidant capacity (DPPH free radical chelating capacity, ferric ion reducing antioxidant capacity, and β-carotene-linoleic acid assay)	[63]
Perilla seed meal	NFF and PMR perilla seed peptides (PSP)	↓ ROS	[64]
Potato	IFGPM, IDGGGI, HGPHIF, VDDDKDFIPF, LVTVDDDKD, VVTGGKVGNEND, potato protein hydrolysates (PPH)	Fe^2+^-chelating activity (55.33 µg EDTA/mL), ·OH scavenging activity (230.05 µg Vc/mL), and oxygen radical absorbance capacity (82.24 µg TE/mL)	[65]
Defatted walnut meal	AASCDQ	AQ exhibited strong antioxidant activity, which demonstrated significant scavenging ability against DPPH (79.40%), superoxide anion (81.00%), and ABTS (67.09%) radicals.	[66]
Cyperus (*Cyperus esculentus* L.)	SFRWQ	↑ SOD, CAT, ↓ ROS, TNF-α, IL-6	[67]
Coix seed (*Coix lacryma-jobi* L.)	FFDR	↑ GSH, CAT, SOD, GSH-Px ↓ GSSG, MDA	[68]
Wheat gluten	LY, PY, YQ, APSY, RGGY	↑ SOD, CAT, GSH-Px, GSH ↓ ROS, MDA	[69]
*Moringa oleifera* leaves	LALPVYN	↑ CAT, GSH-Px, SOD ↓ MDA, ROS	[70]
Soybean	Leu-Ser-Trp (LSW)	Inhibited oxidative stress and reduced superoxide and malondialdehyde levels	[71]
Walnut	HGEPGQQQR VAPFPEVFGK HNVADPQR	Excellent cellular antioxidant activity	[72]

Reactive oxygen species (ROS), superoxide dismutase (SOD), catalase (CAT), glutathione peroxidase (GSH -Px), glutathione (GSH and GSSG), glutathione reductase (GR), 1,1-diphenyl-2-picrylhydrazyl (DPPH), 2,2′-Azinobis-(3-ethylbenzthiazoline-6-sulphonate) (ABTS), oxygen radical absorbance capacity (ORAC).

**Table 3 nutrients-16-03277-t003:** The structure–activity relationship of PDBPs obtained from plants.

Functional Activity	Plant Origins	Molecular Characteristics	Structure–Activity Relationship	Reference
Antioxidant peptides	Pine nut (*Pinus koraiensis* Sieb. et Zucc.)	WYSGK	The presence of Ser increases its beta-pleated sheet content, and the active hydrogen atoms produce a chemical shift.	[134]
Antioxidant peptides	Watermelon seed	RDPEER (P1)	The active site of P1 is located at C_6_H_14_ on Arg. P1 can bind to DPPH/ABTS through hydrogen bonding and hydrophobic interaction.	[135]
Antioxidant peptides	Walnut (*Juglans mandshurica* Maxim.)	Walnut protein hydrolysate (<3 kDa) and peptides KGHLFPN	The tendency of the layer secondary structure to be randomly curled during digestion and the increase in active hydrogen content are favorable conditions for improving its antioxidant capacity.	[136]
Antioxidant peptides	Tartary buckwheat (*Fagopyrum tataricum* (L.) Gaertn)	CTGFVAVR	CR-8 can enhance the antioxidant capacity of damaged cells by interfering with multiple metabolic pathways. This is associated with hydrophobic amino acids, *N*-terminal cysteine (Cys), and others.	[137]
Immunomodulatory peptides	Rice (*Oryza sativa* L.)	11–20 amino acids	S, R, D, E, and T amino acids readily form hydrogen bonds with MHC-II molecules, thereby enhancing innate and adaptive immunity.	[138]
Immunomodulatory peptides	Soybean (*Glycine max* L.)	EKPQQQSSRRGS	EKPQQQQSSRRGS increases phagocytic activity of mouse spleen macrophages and also induces macrophage M1 polarization.	[139]
Immunomodulatory peptides	Sunflower seed (*Helianthus annuus* L.)	MVWGP	MVWGP is the most potent immunomodulatory peptide in all cellular assays, which is attributed to the presence of Met residues.	[140]
Antimicrobial peptides	Genus *Ulva* (Ulvophyceae, Chlorophyta)	HAVY RDRF	HAVY has a large number of hydrogen bonds as an antimicrobial agent and has shown higher antimicrobial efficacy than RDRF in in vitro validation.	[141]
Antimicrobial peptides	*Moringa oleifera*	HVLDTPLL	HVLDTPLL inhibited *S. aureus* growth by interacting with dihydrofolate reductase and DNA gyrase through hydrogen bonding and hydrophobic interactions.	[80]
Antimicrobial peptides	Chia seeds (*Salvia hispanica* L.)	<1 kDa	Significant inhibition effect was reported against *Listeria monocytogenes* for components with molecular weight < 1 kDa.	[142]
Anticancer peptides	Walnuts (*Juglans regia* L.)	<1 kDa	The <1 kDa molecular weight fraction (WPH-M1) exhibited more significant inhibition of HCT116 cell proliferation and induction of apoptosis than other fractions.	[143]
Anticancer peptides	Amaranth seed (*Amaranthus caudutus* L.)	Heat denaturation	Bioactive peptides from amaranth seed protein hydrolysates induced apoptosis and antimigratory effects in breast cancer cells.	[144]
Anticancer peptides	Corn gluten meal	3–5 kDa	The fraction 3–5 kDa effectively inhibited the growth of HepG2 cancer cells.	[145]
Antidiabetic peptides	Highland barley	F-3 (DH-23.86%)	F-3 exhibited strong antiglycation activity, effectively suppressed the non-fluorescent AGE (CML) and the fructosamine level.	[146]
Antidiabetic peptides	Hemp (*Cannabis sativa* L.) seed meal	99% of hydrolysate peptides have molecular weights < 5 kDa, 53.95% are 0.5–1 kDa	The peptides inhibited both intracellular disaccharidase and the transport of glucose at different concentrations.	[147]

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
