# Peer review of "Plant-Derived as Alternatives to Animal-Derived Bioactive Peptides: A Review of the Preparation, Bioactivities, Structure–Activity Relationships, and Applications in Chronic Diseases"

_nutrients, 2024, doi:10.3390/nu16193277_

Round 1
Reviewer 1 Report
Comments and Suggestions for Authors
What are the goals of this review? Please, mention them, clearly, in the abstract.
The type of review (narrative?) should be mentioned in the title, in the abstract, and in the whole manuscript.
What can the readers learn from this study? What are the conclusions, practical implications, and directions for further research? Please, mention this in the abstract.
Lines 28-31: “As consumer demand for protein continues to increase, the intense growth of the livestock industry has led to environmental problems. Animal proteins are no longer able to fully supply people's needs, a situation that has accelerated the search for natural protein ingredients that can replace animal proteins.” – References are missing.
A Methods section should be included. See the example of section 2 of the following paper: https://www.mdpi.com/2304-8158/10/6/1175
Were the figures elaborated by you? You have to mention the copyright.
Section 3.7 should be expanded and discussed in a better way.
Please, discuss your study limitations before the Conclusions section.
Comments on the Quality of English LanguageMinor editing of English language required.
Author Response
We are very fortunate and very honored to have your review. We are very happy to have your professional review. Thank you very much for reviewing our paper and for your valuable suggestions. We appreciate your time and effort in helping us to improve the quality and readability of our thesis. Thank you for every comment you have given us. We have given you our answer after careful consideration. We hope you can feel our sincerity. We have made modifications and detailed replies to your suggestions. Our changes are marked in red on the resubmitted file. Thanks again.
Comments 1: What are the goals of this review? Please, mention them, clearly, in the abstract.
Response 1:
Thank you for pointing that out. We are more than happy to take your advice. This will improve the quality of our articles. We have redescribed this part of the summary to make it clearer and more purposeful. The purpose of this paper is to explore the structure-activity relationship and its application in chronic diseases by summarizing the preparation methods of plant-derived bioactive peptides based on their biological activities. In the future, it is expected to optimize its biological activity, explore the mechanism of action of peptides in vivo and provide a certain basis for designing functional drugs and foods. We have abbreviated and added the purpose and marked it in red.
Comments 2: The type of review (narrative?) should be mentioned in the title, in the abstract, and in the whole manuscript.
Response 2:
We really appreciate your kind reminder. Based on your suggestions, we have changed the title to "Plant-derived as alternatives to animal-derived bioactive peptides: a review of preparation, bioactivities, structure-activity relationships, and applications in chronic diseases ". In addition, we mentioned the types of reviews in the abstract and throughout the manuscript and marked them in red.
Comments 3: What can the readers learn from this study? What are the conclusions, practical implications, and directions for further research? Please, mention this in the abstract.
Response 3:
Thank you very much for your constructive comments. We benefit a lot from reading your suggestions. We have reimagined the description of this section of the abstract to mention conclusions, practical implications, and directions for further research, so that readers can more quickly understand the outline of our review. We have added and marked in red in the manuscript (page 1, line 12-26).
Comments 4: Lines 28-31: “As consumer demand for protein continues to increase, the intense growth of the livestock industry has led to environmental problems. Animal proteins are no longer able to fully supply people's needs, a situation that has accelerated the search for natural protein ingredients that can replace animal proteins.” – References are missing.
Response 4:
Thank you very much for your kind reminder. We have added a reference and marked it in red (page 1, section 1, line 31-33). Thank you for your careful review.
Comments 5: A Methods section should be included. See the example of section 2 of the following paper: https://www.mdpi.com/2304-8158/10/6/1175
Response 5:
We sincerely appreciate your valuable comments. We feel very lucky to have your review, and we found that you were very attentive to help us find the reference. You are truly a talented and kind reviewer and we are honored. Thank you for your reference. We have added the material part in the manuscript and marked it in red (page 2, Section 2, line 75-86).
Comments 6: Were the figures elaborated by you? You have to mention the copyright.
Response 6:
We sincerely appreciate your kind reminder. We understand your concern. We designed these figures ourselves. However, some of the icons in the images do have copyrights, which we have added to the acknowledgements section and marked in red (page 19, line 688).
Comments 7: Section 3.7 should be expanded and discussed in a better way.
Response 7:
Thank you for your valuable feedback on our manuscript. We have expanded and discussed section 3.7 based on your suggestions. We have carefully screened and added several references related to this section. All additions are marked in red (page 14, Section 4.7, line 457-459, 464-470 and 475-480). We believe this will further enhance the readability of our articles. Thanks again.
Comments 8: Please, discuss your study limitations before the Conclusions section.
Response 8:
Thank you for your careful review. Your constructive comments are very helpful to our article. We strongly support your point of view. According to your suggestion, we have added the limitations of the study before the conclusion and marked them in red (page 19, Section 7, line 643-659).
Point 1: Minor editing of English language required.
Response 1:
We truly appreciate your comments. Your comments are kind. We have carefully proofread the revised manuscript to improve the language. We have tried our best to polish the language in the revised manuscript. In addition, thank you again for reviewing our paper and providing valuable suggestions. We are grateful for your support and guidance. We have made the necessary changes according to your comments and believe that the quality of the paper has been greatly improved. If you have any other comments or suggestions, we will be happy to listen and make improvements accordingly.

Reviewer 2 Report
Comments and Suggestions for Authors
Dear Authors,
You have put a lot of work into preparing the material; however, the writing of the paper is not thoroughly thought out. The work has a lot of generalizations. The title directs our attention to the work thoroughly describing the preparation, purification, and characterization of biologically active peptides from plants (PDBPs). The introduction of the paper suggests that it deals with the search for and use of alternative protein sources rather than this pure PDBP. I am not sure if the conclusions of some citations (e.g., L:138-139; 160 or 172) have been correctly interpreted. The text has many factual stumbling blocks, e.g., L237 and MS techniques for purification.
I put just few of them as an example:
L30: meat is also natural and may be an “alternative plant protein” that will better fit
L40-44: Plant protein as an alternative protein to meat has other expectations than, I guess, according to the title, bioactive peptides. Please rewrite this fragment.
L52: It will improve this sentence when you add an example of chronic disease.
L53-55: You do not need to present those data describing the research interest of bioactive peptides; this is not the subject of this manuscript. It will be more informative when you tell why it is so important for human health, for instance, to search for PDBP. The PDBPs are attractive because of their biological activity, not because so many people work on it.
L97: To provide the enzyme amount for the reaction, we use the activity unit per gram of enzyme preparation per gram of protein.
L99: which one?
101: highest than what?
L124-125: If I understand correctly, Ulva sp. is a rich source of PDBP after papain hydrolysis. The PDBPs showed low renin and high ACE-I inhibition activity.
L133: How is it possible to combine one protease hydrolysis? With what?
L138-139: This sentence looks like you read only the title and abstract of the publication to put it into the manuscript.
L160: Nutrient like vit D does not produce any proteases
L162: what does it mean to have good physical and chemical properties?
L172: fermented products or PDBPs?
L237: Do hundreds of amino acids close to protein?
The work requires rethinking the form of the presentation, and I propose resubmitting it after thorough improvement.
Comments on the Quality of English Language
The language is understandable.
Author Response
We would like to thank the reviewers for their valuable time and constructive comments on our articles. We feel your sincerity. We are very glad to have received your review, and your specific suggestions are very helpful to us. After we read your suggestions word by word and thought about them, we carefully revised the article. We are deeply sorry for your mention of writing without thoughtful advice. This may be because our review could really be improved, so we checked and revised it as best we could. We put a lot of effort into preparing this review. We carefully laid out the content framework of the article, and the writing content also referred to and searched a lot of literature. This review is the result of our repeated revisions. I hope you can understand our intention. Regarding the introduction you mentioned that there is no pure PDBPs, we understand your concern. We'll cover how this section has been modified to better highlight PDBPs. In addition, you also mentioned that some of the quotations are correct, we hope you do not worry. Because we read the entire quotation carefully during the writing process, summarize and excerpt it. Our writing also follows the words of the quoted authors, you can rest assured. But there are some quotes that can be ambiguous, which we didn't consider before. Therefore, according to your suggestions, we have deleted and replaced inappropriate citations after repeated confirmation and review of materials. Of course, our description of some citations is not accurate enough, we have checked and changed the full text, and marked it in red. I hope you can feel our sincerity, we have taken your suggestions seriously.
Comments 1: L30: meat is also natural and may be an “alternative plant protein” that will better fit
Response 1:
Thank you very much for your constructive comments. As for your mention of " meat is also natural and may be an “alternative plant protein” that will better fit", we have not notice thoroughly before. We think your suggestion is very reasonable. Therefore, we will recap the introduction section. We shifted the overall background of this review to focus on plant-derived bioactive peptides. We have changed introduction and marked it in red.
Comments 2: L40-44: Plant protein as an alternative protein to meat has other expectations than, I guess, according to the title, bioactive peptides. Please rewrite this fragment.
Response 2:
Thank you for your comments. We fully understand your concern. According to your suggestion, we have rewritten the paragraph and marked it in red (page 1, section 1, line 36-38 and 41-42). Thank you for your sincere advice.
Comments 3: L52: It will improve this sentence when you add an example of chronic disease.
Response 3:
Thank you for pointing this out. We agree with this comment. Therefore, we have added chronic disease examples in the article and made sure that the sentences are smooth. We have added it to the article and marked it in red (page 2, line 54-60).
Comments 4: L53-55: You do not need to present those data describing the research interest of bioactive peptides; this is not the subject of this manuscript. It will be more informative when you tell why it is so important for human health, for instance, to search for PDBP. The PDBPs are attractive because of their biological activity, not because so many people work on it.
Response 4:
Thank you for pointing this out in your manuscript. We very much agree with you. (It will be more informative when you tell why it is so important for human health, for instance, to search for PDBP). The data we provided on the keywords and research interests of plant-derived bioactive peptides can show that plant-derived bioactive peptides are relevant to human health. But you may be worried by our lack of clarity. We rephrase the description of these two graphs and mark them in red in the article (page 2, line 62-70). We will make the following explanation, hope you can understand our ideas. Figure 1A The number of articles related to plant-derived bioactive peptides has been increasing year by year. This can prove that our review has a writing basis and a certain significance. If the number of articles is too small, our summary will be meaningless, and we will not be able to explore the biological activity and other related content. Figure 1B shows the co-occurrence of keywords related to plant-derived bioactive peptides. The results showed that there were many keywords related to human chronic diseases, such as blood pressure, cancer, chronic disease and so on. This proves that there is a correlation between the two and can be reviewed. Therefore, we believe that the data in Figure 1 is meaningful, and we hope you can understand. Thanks again.
Comments 5: L97: To provide the enzyme amount for the reaction, we use the activity unit per gram of enzyme preparation per gram of protein.
Response 5:
Thank you very much for your careful examination. As for the enzyme dose unit you mentioned, we think your suggestion is more professional and reasonable. We also checked the literature cited here again and found that the original content expressed the amount of enzyme as "0.25 g/100 g protein". So, we wonder whether we should follow the description of the original author, we seem to have no right to change some content of the original text.
Comments 6: L99: which one?
Response 6:
We very much appreciate your careful review. According to your suggestion, we have explained this sentence in more detail. This is indeed an oversight on our part. We are sorry that the description is not specific enough, we have added and marked in red in the article (page 3, section 3.1.1, line 111-113).
Comments 7: 101: highest than what?
Response 7:
Thank you for your valuable feedback on our paper. We have modified and marked in red in the article (page 3, section 3.1.1, line 113-116). We fully understand the need to strengthen issues such as these. We have reviewed the whole article and made changes.
Comments 8: L124-125: If I understand correctly, Ulva sp. is a rich source of PDBP after papain hydrolysis. The PDBPs showed low renin and high ACE-I inhibition activity.
Response 8:
Thank you for pointing this out. We agree with this comment and rephrase this sentence. We have made changes in the article and marked them in red (page 4, section 3.1.1, line 139-141). We've also fixed things like that. Thank you again for your constructive comments.
Comments 9: L133: How is it possible to combine one protease hydrolysis? With what?
Response 9:
Thank you very much for your comments. Our representation is not complete enough, we have revised it in the article and marked it in red (page 4, section 3.1.2, line 147-149). We make the following explanation: In simple terms, complex enzymatic hydrolysis uses two or more enzymes to hydrolyze proteins, and this process can be multiple enzymes to hydrolyze proteins one by one, or multiple enzymes to hydrolyze proteins together.
Comments 10: L138-139: This sentence looks like you read only the title and abstract of the publication to put it into the manuscript.
Response 10:
Thank you very much for your comments. We are very sorry for causing such misunderstanding with you. We did read the full article and summarized accordingly. But I'm sorry that our description may not be complete. Therefore, we have made changes in the article and marked them in red (page 4, section 3.1.2, line 152-156).
Comments 11: L160: Nutrient like vit D does not produce any proteases
Response 11:
Thank you very much for your professional comments. We understand your concern. Our relevant discussion is based on literature. But we haven't explored it on a deeper level. After our investigation, we found that specific conditions that require rigor and complexity during microbial fermentation also produce the nutrient vitamin D. So, to avoid ambiguity. We have deleted it.
Comments 12: L162: what does it mean to have good physical and chemical properties?
Response 12:
Thank you very much for your thoughtful comments. Regarding your reference to "L162: what does it mean to have good physical and chemical properties? ". We explain that physical properties are properties of substances that do not require chemical changes, such as color, state, odor, hardness, solubility, etc. Chemical properties are the properties of substances that are manifested in chemical changes. For example, oxidation, reducibility, thermal stability and some other properties. Therefore, these properties are closely related to PDBPs. PDBPs has better physical and chemical properties to better display biological activity.
Comments 13: L172: fermented products or PDBPs?
Response 13:
Thank you very much for your kind reminder. We have verified that the fermentation product in the original paper is the bioactive peptide of plant origin described in this paper. We apologize for the inaccuracy of our description. We've replaced "fermentation product" with "soybean peptides" and labeled them in red (page 5, section 3.1.3, line 186-189).
Comments 14: L237: Do hundreds of amino acids close to protein?
Response 14:
Thank you for pointing this out. We apologize for our oversight. We have made changes in the article and marked them in red (page 6, section 4, line 248-250). Thank you very much for your help and for improving the quality of our articles.
Comments 15: The work requires rethinking the form of the presentation, and I propose resubmitting it after thorough improvement.
Response 15:
Thank you very much for taking time out of your busy schedule to review this manuscript. We are honored to hear your professional opinion. We really appreciate your professional comments, they are very helpful. We have thoroughly revised the manuscript according to your suggestions and responded in detail. Looking forward to your reply.
Point 1: The language is understandable.
Response 1:
Thank you very much for your recognition of our English language. Thank you for every comment you have given us. If there are any other modifications we can make, we would very much like to modify them, and we would also appreciate your help.

Round 2
Reviewer 1 Report
Comments and Suggestions for Authors
Thank you for following my suggestions.